# In-vivo and in-vitro toxicity evaluation of 2,3-dimethylquinoxaline: An antimicrobial found in a traditional herbal medicine

**Abdelbagi Alfadil** [1,2] *, **Hamoud Alsamhan**[3], **Ahmed Ali**[1,3], **Huda Alkreathy**[3], **Mohammad W. Alrabia**[1,2], **Asif Fatani**[1], **Karem A. Ibrahem**[1]

1 Faculty of Medicine, Department of Clinical Microbiology and Immunology, King Abdulaziz University, Jeddah, Saudi Arabia, 2 Center of Research Excellence for Drug Research and Pharmaceutical Industries, King Abdulaziz University, Jeddah, Saudi Arabia, 3 Faculty of Medicine, Department of Pharmacology, King Abdulaziz University, Jeddah, Saudi Arabia

* aegmusa@kau.edu.sa

**Data Availability Statement:** all relevant data are fully included within the manuscript itself.

**Funding:** Grant number IFPIP:1281-140-1443 from the Instructional Improvement Fund

## Abstract

2,3-dimethylquinoxaline (DMQ) is a broad-spectrum antimicrobial phytochemical. This study aims to assess its toxicological profile. *In vitro* studies conducted in appropriate cell cultures, included assessment of cardiotoxicity, nephrotoxicity, and hepatotoxicity. *An in vivo* study was conducted in mice to determine acute oral toxicity (**AOT**), and subacute oral toxicity (**SAOT**). Acute dermal toxicity (**ADT**) was conducted in rats. All *in-vitro* toxicity studies of DMQ had negative results at concentrations ≤100 μM except for a non-significant reduction in the ATP in human hepatocellular carcinoma cell culture. The median lethal dose of DMQ was higher than 2000 mg/kg. All animals survived the scheduled necropsy and none showed any alteration in clinical signs. Biochemistry analysis revealed a significant difference between the satellite and control groups, showing an increase in platelet counts and white blood cell counts by 99.8% and 188.8%, respectively. Histology revealed enlargement of renal corpuscles; hyperplasia of testosterone-secreting cells; and dilatation of coronaries and capillaries. The present data suggests an acceptable safety profile of DMQ in rodents except for thrombocytosis, leukocytosis, and histological changes in high doses that need further investigation.

## 1. Introduction

Charles and Minakiri reported the natural presence of 2,3-dimethylquinoxaline (DMQ) in plants, such as *Chromolaenaodorata* (Fig 1) [1–3]. They suggested that the observed antimicrobial activity of *Chromolaenaodorata* could be attributed to the presence of DMQ. Alfadil and his colleagues confirmed that DMQ had broad-spectrum antimicrobial activity [4].

Few safety studies have described the potential toxicity of DMQ. Beraud et al. reported that DMQ was well tolerated in an oral dose of 500 mg/kg/day in rats with a minor effect on the hepatic enzymes CYP450 [5]. Epler et al. tested DMQ for its mutagenic activity, which then showed negative results using the Ames Salmonella sp. assay [6]. Longstaff reported false

supported this study. We would like to extend our gratitude to the Saudi Arabian Ministry of Education (Moe) and King Abdulaziz University, DSR in Jeddah, for their invaluable technical and financial aid.

**Competing interests:** The authors have declared that no competing interests exist.

positives of the carcinogenic potential of DMQ using a sebaceous-gland-suppression test in mice [7]. Westwood reported the negative carcinogenic potential of DMQ using a tetrazolium-reduction test in mice [8]. Styles confirmed DMQ to be non-carcinogenic using the cell transformation assay in three cell line types [9]. Moriguchi et al. predicted that DMQ is non-carcinogenic using the fuzzy adaptive least-squares (FALS) method [10].

The safety and specific organ toxicity of DMQ is still unknown. Therefore, this study aimed to investigate the in vitro toxicity of DMQ in human embryonic kidney cells (HEK-293), RPTEC, human hepatocellular carcinoma cells (HepG2) and *in vivo* AOT, SAOT and ADT.

## 2. Materials and methods

### Chemicals

Carbonyl cyanide 3-chlorophenylhydrazone (CCCP; Cat. No. C2759), carboxy-methylcellulose (Cat. No. C5678), chlorpromazine (Cat. No. C8138), colistin (Cat. No. C4461), dimethyl-sulfoxide (DMSO; Cat. No. D2650), 2,3-dimethylquinoxaline (DMQ; Cat. No. D184977), N-[4-[1-[2-(6-Methyl-2-pyridinyl)ethyl]-4-piperidinyl]carbonyl] phenyl] methane sulfonamide dihydrochloride (E-4031; Cat. No. M5060) and sertraline (Cat. No. S6319) were bought from Sigma-Aldrich (Dorset, UK).

### 2.2 Cell culture

The human HEK-293, RPTEC and HepG2 cells were obtained from the Public Health England European Collection of Cell Cultures (ECACC, Salisbury, UK) and cultured per the manufacturer's instructions.

### 2.3 Animals

Swiss albino mice and Wistar albino rats were obtained from KFMRC, KAU, Jeddah, SA and certified as healthy animals before conducting the study. Animals were housed in the vivarium

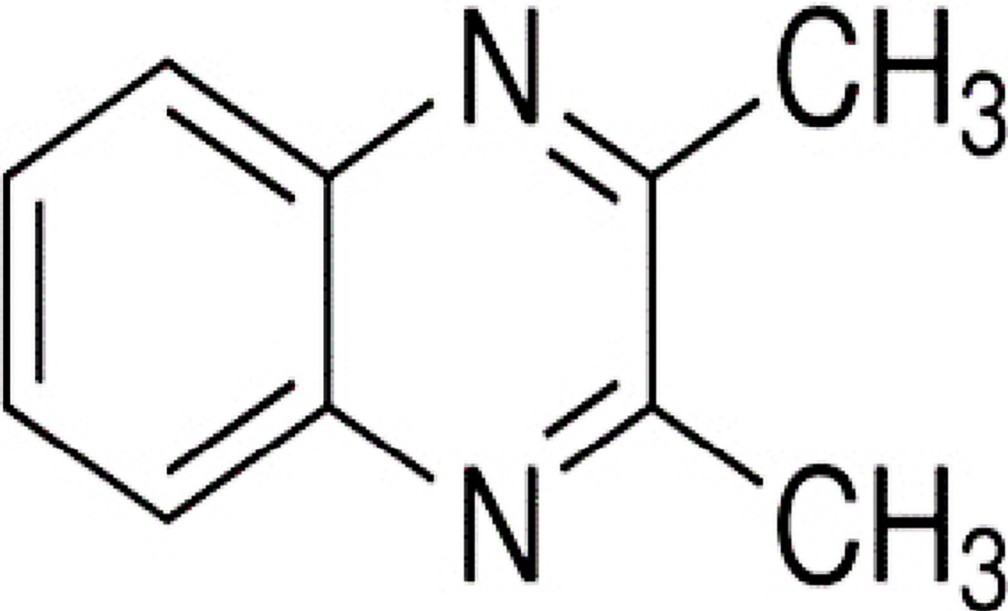

**Fig 1. Structure of 2, 3-dimethyquoxaline.**

of the Faculty of Pharmacy (KAU, Jeddah, SA). Animals were acclimatized for one week before dosing. Environmental conditions were kept at 22±3˚C, 50% humidity along with dark-light cycles (12 hrs.). Animals were fed ad libitum with a freely accessible standard pellet diet and drinking water throughout the experiments. Animals that survived until the experiment's end or reached a moribund state were euthanized humanely. Euthanasia included administering an overdose of ketamine and xylazine intraperitoneally, followed by cervical dislocation. All animals in the AOT (n = 15 mice), ADT (n = 10 rats), and sub-acute repeated 28-day oral toxicity studies (n = 50 mice) underwent this procedure. Post-dosing observational periods lasted up to 28 days for both treatment and control groups, varying based on the experiment's design.

In this study, specific criteria have been employed to evaluate the condition of animals and determine whether euthanasia is warranted. These criteria encompass a range of factors, including the animal's overall health status, behavior, pain levels, distress, and response to treatment or interventions. Additionally, various physiological parameters are monitored, such as body weight, body temperature, respiratory rate, heart rate, and clinical signs of disease or suffering.

It is imperative that decisions regarding euthanasia adhere to established guidelines and ethical principles to minimize animal suffering and ensure humane treatment. These guidelines are typically delineated in institutional animal care and use protocols, which are formulated based on regulatory requirements, scientific norms, and ethical considerations.

This study operates in accordance with the ethical guidelines and research protocols established by the Research Ethics Committee (Faculty of Pharmacy, KAU, Jeddah, SA, Reference No. PH-1443-17).

## 2.4 Ethical approval

Permission to conduct these experiments was taken from the Research Ethics Committee (Faculty of Pharmacy, KAU, Jeddah, SA, Reference No. PH-1443-17). Throughout the trials, the ILAR guideline for the use and care of laboratory animals was adhered to (Council 2010).

## 2.5 Cardiotoxicity assay

The effect of DMQ on the hERG-encoded potassium channel, expressed in HEK-293 cells, was assessed using the Q-Patch HTX electrophysiology whole-cell patch-clamping technique as described previously [11]. DMQ was applied sequentially to the HEK293 cell at four concentrations (0.1, 1, 10 and 25 μM). E-4031 and DMSO were used as positive and negative controls, correspondingly. The cumulative concentration-response curve was obtained. The concentrations that produce 50% inhibition ($IC_{50}$) were calculated and reported as mean ± SD.

## 2.6 Nephrotoxicity assay

The nephrotoxic effects of DMQ were investigated in the RPTEC cell line using a multiparametric high content screening platform (Thermo Scientific Cell omics ArrayScan®) at eight-time points over nine days as described previously [12]. Five cell biomarkers were measured to predict nephrotoxicity risk. These markers include a decrease in glutathione content (GSH, detected by mBCl® Green dye), increase in phospholipids (PLD, detected by Lipid TOXTM® Red dye), increase in DNA fragmentation (detected by Syto11® blue label), mitochondrial dysfunction (detected by Mitro Tracker® Deep Red dye), or decrease in cellular ATP (detected by Cell Titer-Go®). RPTECs were exposed to DMQ for 216 hrs., at 8 concentrations (0, 0.04, 0.1, 0.4, 1, 4 10, 40 and 100 μM) in triplicate, with re-dosing on day 3 and day

6. Colistin and sertraline were utilized as positive controls. The dose-response curves for each biomarker were obtained and reported as mean ± SD.

## 2.7 Hepatotoxicity assay

The hepatotoxicity of DMQ was investigated using HepG2 cell viability assay based on the quantification of ATP as described earlier [13]. Cells were seeded inside a 96-well plate (Falcon), adjusted to $5 \times 10^4$ cells per well, and cultured at 37 ˚C in 5% $CO_2$ for 24 hrs. and 95% air atmosphere. After incubation, cells were treated with DMQ in 8 concentrations (0.04, 0.1, 0.4, 1, 4, 10, 40 and 100 μM) with 3 replications per concentration. Chlorpromazine and carbonyl cyanide 3-chlorophenylhydrazone were used as positive controls. Treated cells were incubated for a further 72 hrs. The final DMSO concentration was less than 0.5%. At the end of the incubation period, Cell Titer-Glo® Buffer and Substrate were mixed per the manufacturer's instructions (Promega, Southampton, UK), and then added 100 μl to each well containing the treated cells. The plates' contents were gently mixed using an orbital shaker at room temperature for 10 mins. Luminescence was recorded using a luminometer (BMG Labtech, Aylesbury, UK). The concentration at which $AC_{50}$ is seen for ATP cell health parameters was reported as mean ± SD.

## 2.8 Acute oral toxicity study in mice

AOT of the DMQ study in mice was conducted at the vivarium of the Faculty of Pharmacy following the OECD guideline 423 for testing chemicals [14]. Swiss albino mice ($n$ = 15, female, 6–8 weeks old) were used for this research. Mice in the control group ($n$ = 3) were given a 10 ml/kg carboxymethyl cellulose (0.5% CMC) vehicle only via oral gavage. In the treatment groups, mice (n = 3 per group) were given DMQ suspended in the vehicle via oral gavage. Treatment groups were begun in ascending steps (5, 50, 300 and 2000 mg/kg). If no mice died in the lower dose group, we went ahead with the higher dose group. All mice were seen at 0.5, 1, 2, 4 and 24 hrs. post-dosing on day one and daily after that for a total of 14 days. These observations included locomotion, lacrimation, salivation, sniffing, defecation, catatonia, convulsive, piloerection, visual place and tail pinch response.

## 2.9 Oral toxicity study with repeated doses in mice (28 days)

SAOT of the DMQ study in mice was conducted at the vivarium of the Faculty of Pharmacy following OECD Guideline number 407 for Testing Chemicals [15]. Swiss albino mice ($n$ = 50, 6–8 weeks old) were used for this study. Five groups of ten mice each were randomly assigned; vehicle control (10 ml/kg CMC 0.5%), low dose (250 mg/kg/day DMQ), medium dose (500 mg/kg/day DMQ), high dose (1000 mg/kg/day DMQ) and satellite group (1000 mg/kg/day DMQ with an additional 14 days post-dosing observational period). The treatments and the control were given via oral gavage, daily, at the same time, for 28 days. Twice daily examinations were made throughout the study. Weekly measurements were made for body weight and feed consumption. Before necropsy, blood samples were obtained from the sedated mice using their retro-orbital. Various biochemical and hematological parameters were investigated for all the mice. Vital organs were isolated, and their absolute and relative weights (i.e divided by body weight) were determined. Samples of the organs were fixed in 10% formalin and tissue-paraffin embedding blocks were made. These blocks were processed using a microtome at 5-μm thickness (Leica, Germany), stained with hematoxylin and eosin (H and E), and examined via a light microscope (Nikon, Eclipse 80i, Japan).

## 2.10 Acute dermal toxicity study in rats

ADT of the DMQ study in rats was conducted at the vivarium of the faculty of pharmacy following OECD Guideline 434 for Testing Chemicals [16]. Wistar albino rats ($n$ = 10, female, non-pregnant, 12–14 weeks old) were used in this research. Rats were then randomly divided into five rats in the control group and five in the treatment group. The DMQ dose was 2000 mg/kg, which is the upper-limited dose in the protocol. Their fur was carefully shaved from the dorsal area (10% of the body surface area) 24 hours before dosing. The treatment was applied locally once on day 1 of the study. All rats were seen at 0.5, 1, 2, 4 and 24 hrs. post-dosing on day one and daily after that for a total of 14 days. These observations included locomotion, lacrimation, salivation, sniffing, defection, catatonia, convulsive, piloerection, visual place and tail pinch response.

## 2.11 Data analysis

ANOVA and Dunnett's tests were applied to find statistically significant differences. Data analysis was performed by GraphPad Prism 9.3.1. Data was expressed as mean ±SD with a $P<0.05$ significance level.

# 3. Results

## 3.1 Cardiotoxicity of 2,3-dimethylqinoxline *in-vitro*

DMQ and DMSO showed an absence of *in-vitro* hERG channel inhibition activity at concentrations ≤25 μM. Whereas, E-4031 showed a strong inhibition of the hERG channel with $IC_{50}$ = 0.03 μM and a peak activity $pIC_{50}$ = 7.54 ±0.03 μM.

## 3.2 Nephrotoxicity of 2,3-Dimethylqinoxline *in-vitro*

DMQ, colistin and sertraline showed no significant effect on RPTEC cell count, no necrosis, apoptosis or cellular proliferation reduction at concentrations ≤100 μM over 216 hrs. incubation time. Only colistin showed an increase in nuclear size ($EC_{50}>$ 200 μM), indicating G2 cell cycle arrest. Colistin and sertraline showed an increase in DNA structure ($EC_{50}$ = 64.8 μM and 16.4 μM, respectively), indicating chromosomal instability and DNA fragmentation. Also, colistin and sertraline showed a decrease in mitochondrial mass ($IC_{50}$ = 67.3 μM and 10.2 μM, respectively) and an increase in mitochondrial membrane potential (($EC_{50}>$ 120 μM and 17.6 μM, respectively), showing an adaptive response to cellular energy demands. Colistin and sertraline showed an increase in phospholipids ($EC_{50}$ = 63.3 μM and 6.47 μM, respectively), showing an accumulation of phospholipids and lysosomes. Colistin and sertraline showed a decrease in glutathione ($IC_{50}$ = 84 μM and 2.41 μM, respectively), showing a cellular response to oxidative stress. Only colistin and sertraline showed a decrease in ATP ($IC_{50}$ = 33.1 μM and 9.13 μM, respectively), indicating a decrease in metabolically active cells (Fig 2).

## 3.3 Hepatotoxicity of 2,3-dimethyl quinoxaline *in-vitro*

DMQ showed a nonsignificant reduction in HepG2 cell ATP with $IC_{50}$ being more than 100 μM. Chlorpromazine and carbonyl cyanide 3-chlrphenylhydrazone showed a significant reduction in ATP ($IC_{50}$ = 10.5 and 2.67 μM, respectively), indicating a decrease in metabolically active and healthy HepG2 cells after exposure to these compounds (Fig 3).

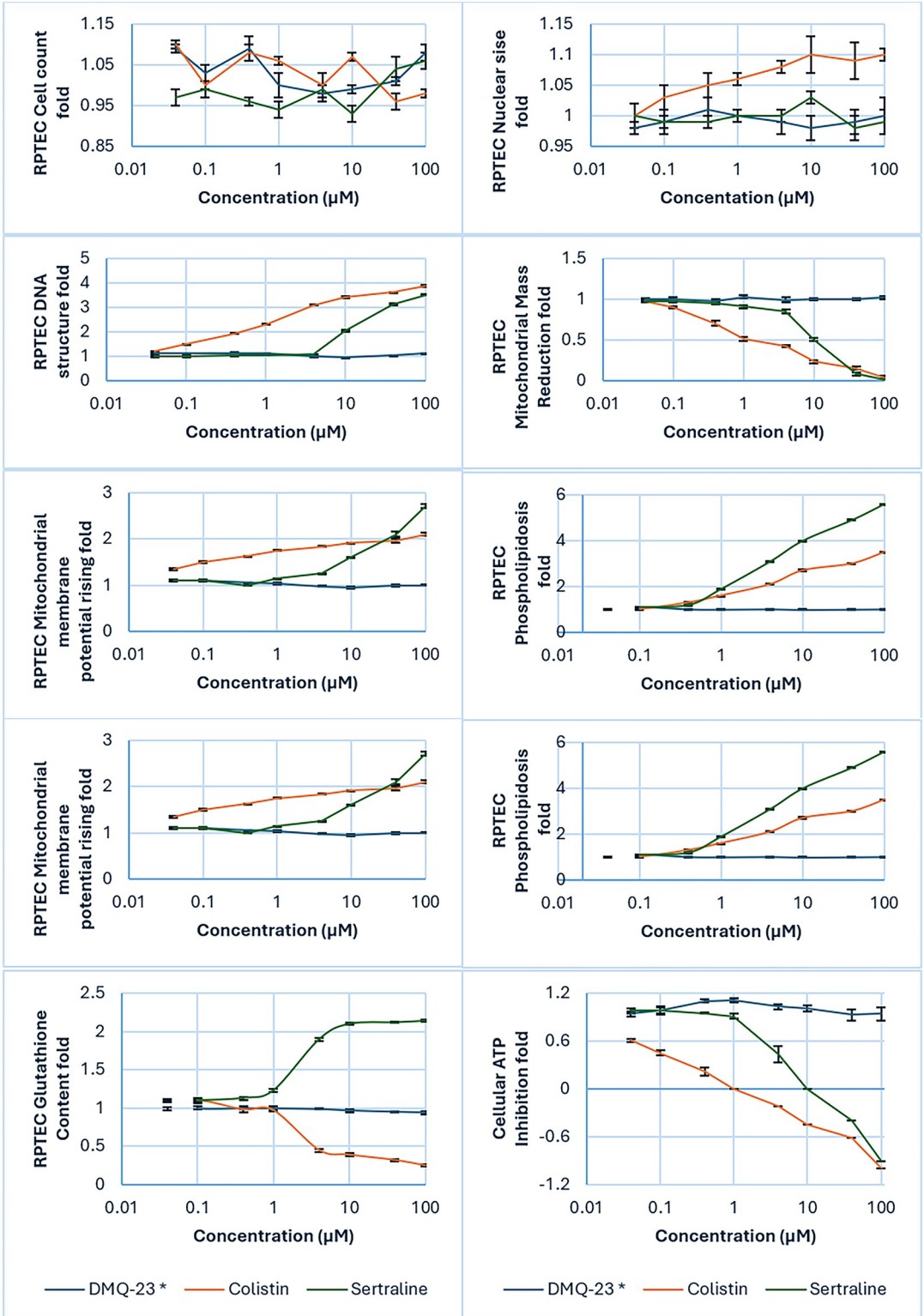

**Fig 2. *In-vitro* nephrotoxicity study of 2,3-dimethylquinoxlaline (DMQ-23) compared to colistin and sertraline in RPETC cell line using eight parametric endpoints.**

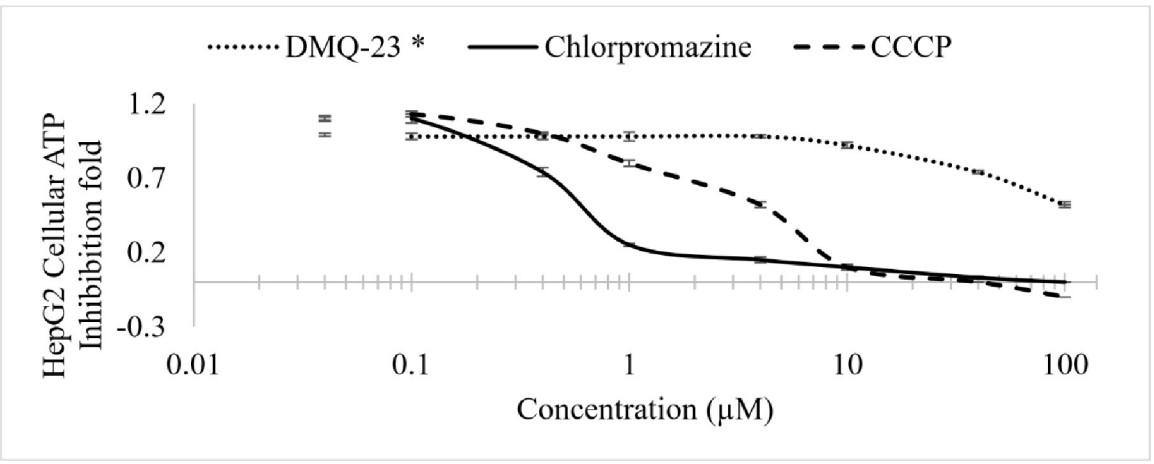

**Fig 3. *In-vitro* hepatotoxicity study of 2,3-dimethylquinoxaline (DMQ-23) compared to chlorpromazine and carbonyl cyanide 3-chlorphenylhydrazone in HepG2 cell line using cellular ATP as a parametric endpoint.**

### 3.4 Animal toxicity studies

**3.4.1 Mortality and morbidity.** All animals in the AOT ($n = 15$ mice), ADT ($n = 10$ rats) and sub-acute repeated 28-day oral toxicity studies ($n = 50$ mice) survived scheduled necropsy. None of these animals showed any alteration in clinical signs of secretary, muscularity, or reflexive activity during the observation period.

**3.4.2 Effect on food and water intake.** The food and water consumption of all the animals in the AOT ($n = 15$ mice) and ADT ($n = 10$ rats) experiments did not differ significantly. In contrast to the control group, the high-dosage and satellite groups in the repeated dose 28-day oral toxicity study had a significant ($P<0.05$) increase in food intake (Table 1).

**3.4.3 Effect on body weight and organ weight.** In both AOT ($n = 15$ mice) and ADT ($n = 10$ rats) trials, there was no discernible variation in the body and organ weight of any of the animals. In the 28-day oral toxicity trial with repeated doses, the low-dose and high-dose groups saw a significant ($P<0.05$) decrease in body weight in comparison to the control groups (Table 1).

**3.4.4. Hematology analysis.** A subacute repeated 28-day oral toxicity trial revealed substantial differences ($P<0.05$) in several haematological parameters between the treatment groups and the control group at necropsy. Table 2 shows a statistically significant ($P<0.05$) decrease in haemoglobin and red blood cell counts and a statistically significant rise in platelet and white blood cell counts.

**3.4.5. Biochemistry analysis.** At necropsy, some of the biochemistry parameters in the subacute repeated 28-day oral toxicity study revealed significant differences ($P<0.05$) between the satellite and control groups in liver function, kidney function, triglycerides and cardiac biomarkers (Table 3).

**3.4.6 Histopathological analysis.** In the subacute repeated 28-day oral toxicity study, there were no significant findings in internal or external gross pathology. Absolute and relative organ weights did not show significant differences (Table 1). No other alteration was seen in the splenic red pulp or the marginal zone. Kidney histopathology revealed the enlargement of renal corpuscles with no changes in glomerular capillaries and a few dilated sporadic tubules. Testis histopathology revealed normal seminiferous tubules (Fig 4), mild blood vessel dilatation with congestion and hyperplasia of testosterone-secreting cells (Fig 5). Heart histopathology revealed normal histology except for coronaries 'dilatation with increased perivascular

**Table 1. Effect of 28-days oral dosing of 2,3-Dimethylquinoxaline on the physical parameter in Swiss albino mice.**

| | Control group | DMQ-23 treatment groups | | | |
| --- | --- | --- | --- | --- | --- |
| | | Low-dose | Medium-dose | High-dose | Satellite |
| **Food Intake** (g/day) | | | | | |
| | 7.2 ± 0.57 | 7.1 ± 0.38 | 6.7 ± 0.49 | 8.1 ± 0.32 * | 7.8 ± 0.45 * |
| **Water Intake** (ml/day) | | | | | |
| | 8.5 ± 0.31 | 8.6 ± 0.4 | 8.3 ± 0.52 | 8.2 ± 0.34 | 8.2 ± 0.56 |
| **Bodyweight** (g) | | | | | |
| 1st week | 26.2 ± 2.5 | 25.7 ± 2.3 | 26.1 ± 2.11 | 26.8 ± 2.2 | 25.9 ± 2.8 |
| 2nd week | 26.43 ± 2.8 | 25.81 ± 2 | 26.22 ± 2.6 | 27.12 ± 3.2 | 26.47 ± 1.8 |
| 3rd week | 26.92 ± 2.6 | 26.3 ± 2.7 | 26.91 ± 2.2 | 27.43 ± 2.3 | 26.92 ± 2.5 |
| 4th week | 27.4 ±2.47 | 26.66 ± 2.32 | 27.23 ± 2.14 | 27.78 ± 2.52 | 27.2 ± 2.73 |
| **Bodyweight gain** (g) | | | | | |
| | 1.2 ± 0.03 | 0.96 ± 0.02 * | 1.13 ± 0.03 | 0.98 ± 0.32 * | 1.3 ± 0.07 |
| **Organ ratio to the body weight** (%) | | | | | |
| **Heart** | 0.48 ± 0.03 | 0.43 ± 0.05 | 0.51 ± 0.02 | 0.49 ± 0.07 | 0.44 ± 0.04 |
| **Kidney** | 0.63 ± 0.07 | 0.69 ± 0.02 | 0.62 ± 0.04 | 0.64 ± 0.04 | 0.61 ± 0.03 |
| **Liver** | 5.2 ± 0.17 | 5.19 ± 0.09 | 4.91 ± 0.12 | 4.53 ± 0.15 | 5.1 ± 0.11 |
| **Spleen** | 0.36 ± 0.03 | 0.45 ± 0.05 | 0.39 ± 0.08 | 0.41 ± 0.04 | 0.39 ± 0.04 |
| **Brain** | 1.1 ± 0.12 | 1.08 ± 0.06 | 1.17 ± 0.09 | 1.13 ± 0.07 | 1.11 ± 0.12 |
| **Lung** | 0.84 ± 0.07 | 0.83 ± 0.08 | 0.88 ± 0.02 | 0.85 ± 0.06 | 0.81 ± 0.04 |

Data are presented as mean ± S.D. One-Way ANOVA and Dunnett test * Significant at $P<0.05$ $n = 10$.

mast cell, prominence fibroblasts active nuclei and dilated capillaries (Fig 6). Brain, lungs and spleen histology showed typical architectures.

## 4. Discussion

It is encouraging to know that several quinoxaline derivatives successfully passed all safety requirements and were approved for use in humans for several therapeutic indications [17,18]. Quinoxaline derivatives are distinguished by their diverse therapeutic potentials and relatively simple chemical synthesis; therefore, they occupy a prominent place in medicinal chemistry [19].

**Table 2. Hematological parameters of 2,3-dimethylquinoxaline subacute oral toxicity in Swiss albino mice at necropsy.**

| | Control group | DMQ-23 treatment groups | | | |
| --- | --- | --- | --- | --- | --- |
| | | Low-dose 250 mg/kg/d | Medium-dose 500 mg/kg/d | High-dose 1 g/kg/d | Satellite 1 g/kg/d + 14d |
| **RBC** (x10^6/mm^3) | 5.51 ± 0.64 | 3.8 ± 0.34 * | 5.8 ± 2.02 | 3.5 ± 0.37 * | 2.7 ± 0.41 * |
| **Hb** (g/dL) | 13.67 ± 1.02 | 10.36 ± 0.9 * | 13.1 ± 3.23 | 10.8 ± 0.67 * | 10.6 ± 0.8 * |
| **PLT** (cell/mm^3) | 572 ± 160 | 992 ± 176 * | 977 ± 145 * | 1085 ± 126 * | 1143 ± 60 * |
| **WBC** (x10^3/mm^3) | 5.4 ± 0.61 | 10.5 ± 1.04 * | 9.04 ± 2.06 * | 12.4 ± 1.2 * | 15.6 ± 1.9 * |
| NEUT (%) | 12.8 ± 0.31 | 14.3 ± 0.92 | 21.7 ± 1.43 * | 25.6 ± 2.39 * | 28 ± 3.7 * |
| LYMPH (%) | 86.3 ± 7.61 | 71.8 ± 2.83 * | 68.9 ± 5.71 * | 63.8 ± 7.23 * | 51 ± 2.6 * |
| MONO (%) | 0.7 ± 0.03 | 0.9 ± 0.03 | 4.2 ± 0.58 * | 6.2 ± 0.79 * | 16.3 ± 1.5 * |
| EO (%) | 0.03 ± 0.01 | 2.1 ± 0.46 * | 3.2 ± 0.61 * | 3.4 ± 0.52 * | 4.2 ± 0.8 * |
| BASO (%) | 0.00 ± 0.00 | 0.02 ± 0.001 | 0.13 ± 0.02 | 0.22 ± 0.04 | 0.67 ± 0.6 * |

Data are presented as mean ± S.D. One-Way ANOVA and Dunnett test * Significant at $P<0.05$ $n = 10$.

**Table 3. Serum biochemistry parameters of 2,3-dimethylquinoxaline repeated-dose 28-days oral toxicity in mice at necropsy.**

| | Control group | DMQ-23 treatment groups | | | |
|---|---|---|---|---|---|
| | | Low-dose 250 mg/kg/d | Medium-dose 500 mg/kg/d | High-dose 1 g/kg/d | Satellite 1 g/kg/d + 14d |
| **Liver function biomarkers** | | | | | |
| ALT (U/L) | 79.4 ± 14.64 | 98.7 ± 23.13 | 98.3 ± 29.4 | 98.3 ± 16.59 | 114.8 ± 38.28 * |
| ALP (U/L) | 57 ± 10.61 | 58.9 ± 8.45 | 69.7 ± 6.55 | 75.2 ± 24.98 * | 121 ± 20.89 * |
| AST (U/L) | 154.2 ± 28.5 | 145.8 ± 9.11 | 121.5 ± 19.23 | 144.4 ± 40.11 | 344.4 ± 106.8 * |
| Alb (g/L) | 28.34 ± 0.25 | 29.14 ± 0.87 | 28.21 ± 0.86 | 27.88 ± 2.87 | 28.95 ± 0.46 |
| **Cardiac biomarkers** | | | | | |
| CK (U/L) | 297.8 ± 75.9 | 312.3 ± 59.3 | 334.2 ± 63.2 | 393.4 ± 44 * | 2147 ± 3.75 * |
| **Kidney function biomarkers** | | | | | |
| BUN (mmol/L) | 8.71 ± 0.48 | 8.24 ± 0.21 | 7.15 ± 1.17 * | 9.32 ± 1.08 | 7. 1 ± 0.21 * |
| Crea (μmol/L) | 13.4 ± 0.52 | 13.8 ± 1.03 | 14.2 ± 1.55 | 13 ± 0.52 | 19.2 ± 3.39 * |
| **Electrolytes** | | | | | |
| Na (mmol/L) | 147 ± 0.82 | 148.6 ± 0.52 | 149.4 ± 2.07 | 151.9 ± 4.83 * | 144.7 ± 1.42 |
| K (mmol/L) | 5.37 ± 0.12 | 4.56 ± 0.21 | 4.58 ± 0.44 | 4.94 ± 0.47 | 5.24 ± 0.63 |
| Cl (mmol/L) | 112.6 ± 1.71 | 110.7 ± 3.1 | 112.4 ± 0.97 | 115.9 ± 4.23 | 112.5 ± 0.53 |
| Mg (mmol/L) | 0.95 ± 0.04 | 0.98 ± 0.03 | 0.97 ± 0.06 | 1.02 ± 0.11 | 0.84 ± 0.06 |
| Ca (mmol/L) | 2.14 ± 0.05 | 2.18 ± 0.07 | 2.25 ± 0.15 | 2.25 ± 0.18 | 2.15 ± 0.08 |
| P (mmol/L) | 2.35 ± 0.11 | 2.22 ± 0.26 | 2.65 ± 0.36 | 2.41 ± 0.39 | 2.44 ± 0.23 |
| **Metabolites biomarkers** | | | | | |
| Chol (mmol/L) | 2.37 ± 0.34 | 2.29 ± 0.4 | 2.51 ± 0.09 | 2.43 ± 0.71 | 1.87 ± 0.47 |
| LDL (mmol/L) | 0.18 ± 0.04 | 0.18 ± 0.02 | 0.14 ± 0.01 | 0.16 ± 0.03 | 0.13 ± 0.01 |
| HDL (mmol/L) | 0.97 ± 0.15 | 1.26 ± 0.46 | 1.1 ± 0.13 | 1.09 ± 0.34 | 0.78 ± 0.22 |
| Trig (mmol/L) | 0.95 ± 0.2 | 1.35 ± 0.25 | 1.13 ± 0.46 | 1.02 ± 0.27 | 1.76 ± 0.51 * |
| TP (g/L) | 50.9 ± 3.75 | 52.6 ± 0.97 | 48.2 ± 0.92 | 51.4 ± 4.35 | 49.5 ± 0.53 |
| UrcA (μmol/L) | 332.1 ± 31.19 | 334 ± 20.66 | 226.4 ± 36.22 | 163.3 ± 51.66 | 229.5 ± 19.5 |

Values are expressed as mean ± SD of 10 mice. * Significant at the level of $P < 0.05$.

DMQ is a non-teratogenic, non-carcinogenic antimicrobial quinoxaline derivative with herbal medicinal origin. DMQ showed promising biological activity and safety profile, which makes it attractive for further development. The present study is novel regarding basic data on the safety of DMQ. It showed a favourable *in-vitro* safety profile. The study also represents the 1st OECD-based toxicity of DMQ in rodents. DMQ was well tolerated in mice and rats, and no adverse effects were seen. This finding supports what Beraud et al. reported in 1975 [5].

The peripheral WBC count rises in the DMQ treatment groups correlated with the spleen histology study finding, showing partial storage pool depletion and lymphocyte mobilization. This effect on WBC is an added advantage in achieving the treatment goal of controlling infections and invading microorganisms. Further animal study is needed to understand and document DMQ-induced leukocytosis and its short- and long-term effects.

Thrombocytosis was the second main haematological finding. No earlier information about DMQ-induced thrombocytosis has been found in the literature. Signs and symptoms of the ischemic event were not detected throughout the observation period. Further animal study is needed to understand and document DMQ-induced thrombocytosis and its short- and long-term effects. The release of inflammatory cells and cellular response to infection could be hypothetical explanations for the occurrence of secondary thrombocytosis [20].

All the changes in heart histology after exposure to high dose DMQ for 28 days were mild but worth further investigation and confirmation. One explanation for the dilation of

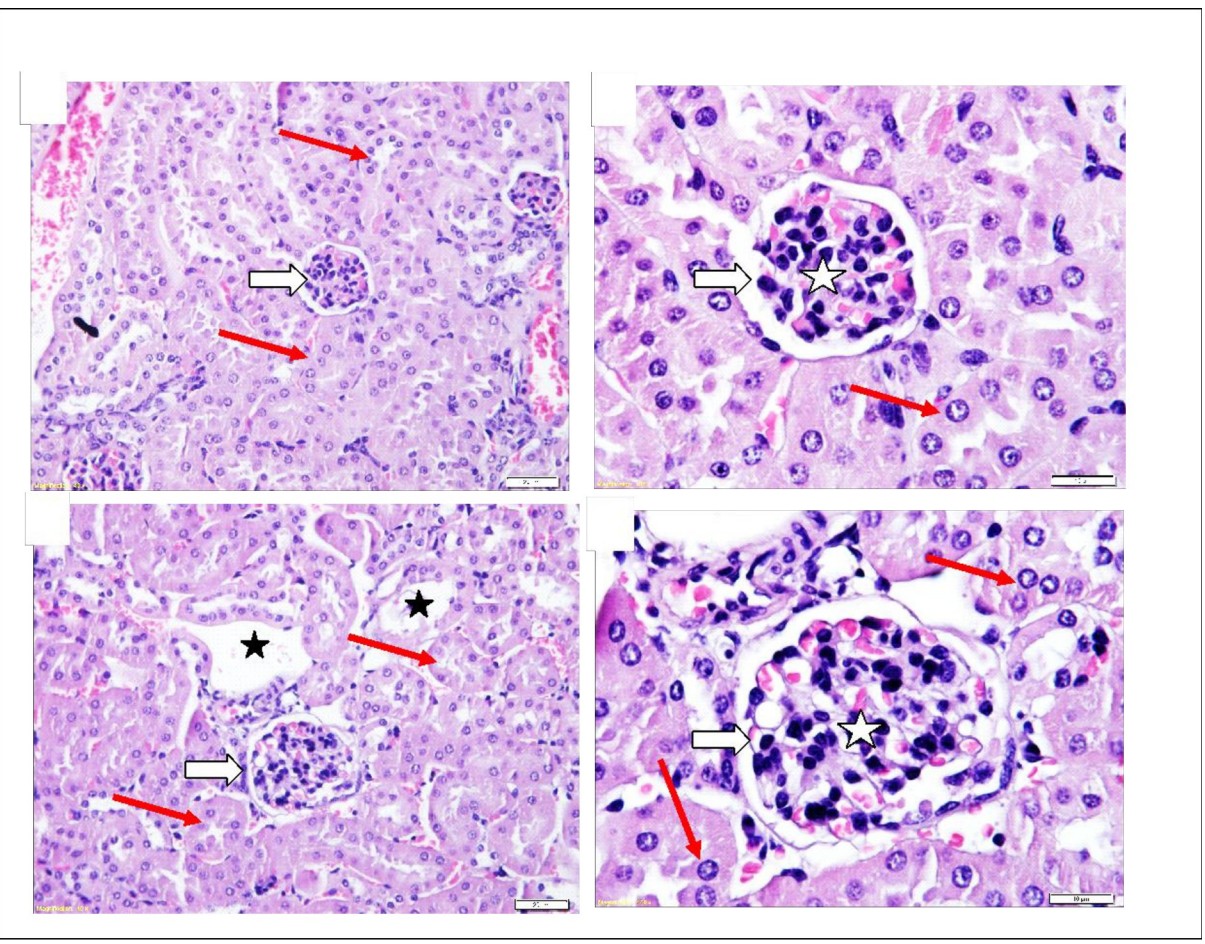

**Fig 4.** Histopathology of mice kidney in subacute Quinoxaline-2,3-Dimethyl oral toxicity study revealed: (A and B) Control group showing a typical architecture of renal corpuscle (white arrows), glomerular capillaries (white star), renal tubules (black arrows), and distal tubules (black star). (C and D) High-dose group (QUI-2 at 1000 mg/kg/day daily dosing for 28 days) showed enlarged renal corpuscle (white arrows) with a slight increase in cellularity of glomerular capillaries (white star). Renal tubules are normal (black arrows) lines with cuboidal intact epithelium with rounded vascular active nuclei (red arrows) and few dilated distal tubules (black star). H&E Stained.

coronaries and capillaries findings could be endothelial nitric oxide (NO) release [21]. It has been reported that 49% of mast cells in mice heart histology exist in the perivascular. The rising mast cells in heart tissues may be linked to various cardiovascular diseases [22]. It is worth more investigation for the increase in mast cells in perivascular tissue. Cardiac fibroblasts were intact after 28 days of exposure to DMQ. Most cardiac activities are mediated by cardiac fibroblasts, the major cell type in the heart [23].

Kidney histology showed minimal change with mild enlargement of renal corpuscles and a few dilated sporadic tubules, which could be supported by biochemistry findings [24].

Unfortunately, testosterone levels were not measured in the subacute toxicity studies. Therefore, it is hard to correlate any goal finding with the testis histology finding of normal seminiferous tubules, mild blood vessel dilatation with congestion and hyperplasia of testosterone-secreting cells. These histological findings could explain the weight gain directly or indirectly, which is worth further investigation

Preclinical research is essential to supply solid data supporting the safety and efficacy of protecting human participants in clinical trials and beyond. Preclinical *in-vitro* studies have

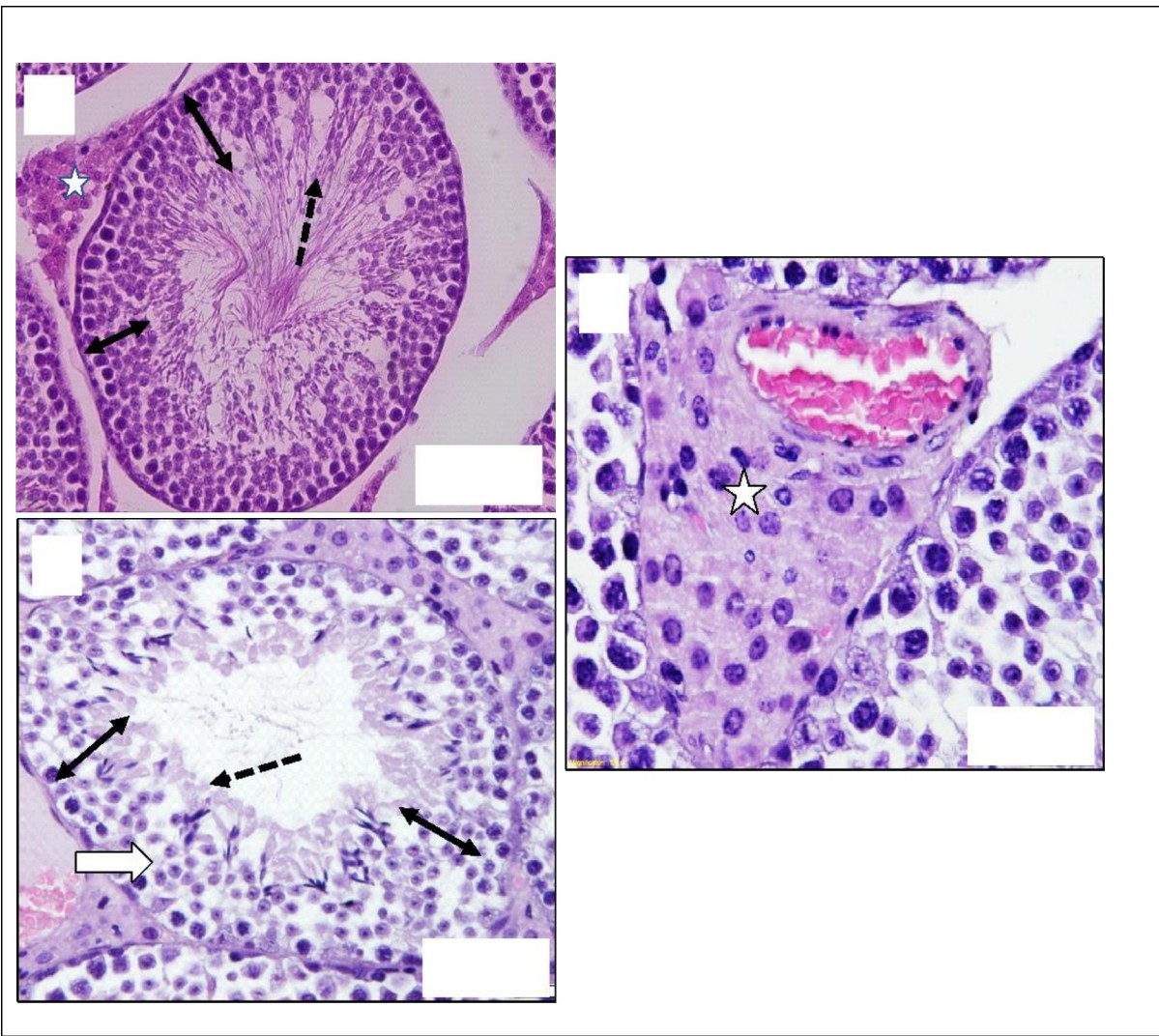

**Fig 5.** Histopathology of mice testis in subacute Quinoxaline-2,3-Dimethyl oral toxicity study revealed: (A) control group showing a typical architecture of seminiferous tubules (ST), germ cell layers (Double head arrows), sperm heads (Black arrows) and interstitial cells (White Star). (B and C) High-dose group (QUI-2 at 1000 mg/kg/day daily dosing for 28 days) showing hyperplasia of interstitial cells (White Star) and congested blood vessels (V) between tubules. Also, it shows normal seminiferous tubules with full-thickness germ cell layers (Double head arrows) except for a few sperm heads (Black arrows) without apparent tails. H&E Stained.

both advantages and disadvantages in exploring specific properties without interfering with body organ systems or other cellular factors. However, due to species variability, there are some limitations. So, data from preclinical *in-vitro* and *in-vivo* experiments should always be taken with precautions when extrapolating the results from preclinical models to the clinical situation [25,26].

## 5. Conclusion

The cumulative results support the safety of DMQ in rodents and cell culture experiments. These also support the continued development of DMQ and its derivatives to further progress into the antimicrobial pipeline. Future toxicity studies are needed in non-rodent species to enhance the generalizability of these findings across species.

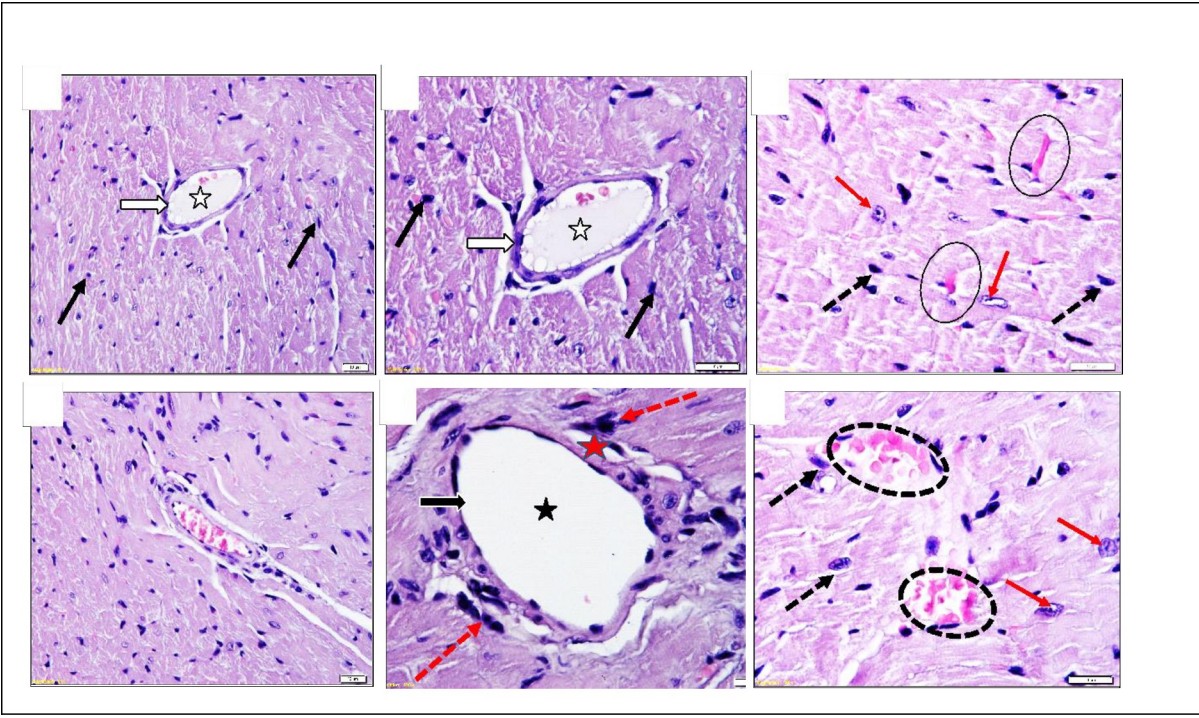

**Fig 6.** Histopathology of mice heart in subacute Quinoxaline-2,3-Dimethyl oral toxicity study revealed: (A, B, and C) Control group showing a typical architecture of coronary artery (white arrows), lumen (white star), capillaries (Black circles), and cardiac fibers (red arrows). (D, E, and F) High-dose group (QUI-2 at 1000 mg/kg/day daily dosing for 28 days) showed dilated congested capillaries (dotted circles) and prominent fibroblast nuclei (dotted black arrows). The perivascular area (red star) contains numerous mast cells (red dotted arrows) and fibroblast nuclei (black arrows). Sections show branches of the coronary artery (white arrows) with a wide lumen (white star), thin-wall non-congested capillaries (Black circles), and active vascular nuclei of cardiac fibers (red arrows). H&E Stained.

## Author Contributions

**Conceptualization:** Hamoud Alsamhan, Huda Alkreathy, Asif Fatani.

**Data curation:** Abdelbagi Alfadil, Hamoud Alsamhan, Karem A. Ibrahem.

**Formal analysis:** Ahmed Ali, Asif Fatani.

**Funding acquisition:** Abdelbagi Alfadil, Karem A. Ibrahem.

**Investigation:** Ahmed Ali, Asif Fatani.

**Methodology:** Abdelbagi Alfadil, Hamoud Alsamhan, Karem A. Ibrahem.

**Project administration:** Ahmed Ali, Huda Alkreathy, Mohammad W. Alrabia, Asif Fatani.

**Resources:** Ahmed Ali, Huda Alkreathy, Mohammad W. Alrabia, Karem A. Ibrahem.

**Software:** Abdelbagi Alfadil, Ahmed Ali, Huda Alkreathy, Mohammad W. Alrabia, Asif Fatani.

**Supervision:** Abdelbagi Alfadil, Huda Alkreathy, Mohammad W. Alrabia, Karem A. Ibrahem.

**Validation:** Huda Alkreathy, Mohammad W. Alrabia, Asif Fatani.

**Visualization:** Hamoud Alsamhan, Ahmed Ali, Huda Alkreathy, Karem A. Ibrahem.

**Writing – original draft:** Abdelbagi Alfadil, Hamoud Alsamhan, Huda Alkreathy, Asif Fatani.

**Writing – review & editing:** Ahmed Ali, Mohammad W. Alrabia, Karem A. Ibrahem.

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
