## [Decision Letter · Decision Letter 0]

4 Apr 2024

PONE-D-24-06527In-Vivo and In-Vitro Toxicity Evaluation of 2,3-Dimethylquinoxaline: an Antimicrobial Found in a Traditional Herbal MedicinePLOS ONE

Dear Dr. Alfadil,

Thank you for submitting your manuscript to PLOS ONE. After careful consideration, we feel that it has merit but does not fully meet PLOS ONE’s publication criteria as it currently stands. Therefore, we invite you to submit a revised version of the manuscript that addresses the points raised during the review process.

We look forward to receiving your revised manuscript.

Kind regards,

Ahmed E. Abdel Moneim

Academic Editor

PLOS ONE

3. To comply with PLOS ONE submissions requirements, in your Methods section, please provide additional information regarding the experiments involving animals and ensure you have included details on (1) methods of sacrifice, (2) methods of anesthesia and/or analgesia, and (3) efforts to alleviate suffering.

“Grant number IFPIP:1199-140-1443- 260449 from the Instructional Improvement Fund supported this study.  Saudi Arabian Moe and King Abdulaziz University, DSR in Jeddah.”

6. Please provide a complete Data Availability Statement in the submission form, ensuring you include all necessary access information or a reason for why you are unable to make your data freely accessible. If your research concerns only data provided within your submission, please write "All data are in the manuscript and/or supporting information files" as your Data Availability Statement.

7. Thank you for stating the following in the Acknowledgments Section of your manuscript:

“Grant number IFPIP:1281-140-1443 from the Instructional Improvement Fund supported this study. The authors would like to express their appreciation to the Saudi Arabian Moe and King Abdulaziz University, DSR in Jeddah for their technical and financial aid”

“Grant number IFPIP:1199-140-1443- 260449 from the Instructional Improvement Fund supported this study.  Saudi Arabian Moe and King Abdulaziz University, DSR in Jeddah.”

8. Please include a separate caption for each figure in your manuscript.

9. Please include your tables as part of your main manuscript and remove the individual files. Please note that supplementary tables should be uploaded as separate "supporting information" files

Reviewers' comments:

Reviewer's Responses to Questions

**Comments to the Author**

1. Is the manuscript technically sound, and do the data support the conclusions?

Reviewer #1: Yes

Reviewer #2: Yes

2. Has the statistical analysis been performed appropriately and rigorously? 

Reviewer #1: Yes

Reviewer #2: Yes

3. Have the authors made all data underlying the findings in their manuscript fully available?

Reviewer #1: Yes

Reviewer #2: Yes

4. Is the manuscript presented in an intelligible fashion and written in standard English?

Reviewer #1: Yes

Reviewer #2: Yes

5. Review Comments to the Author

Reviewer #1: The manuscript entitled “In-Vivo and In-Vitro Toxicity Evaluation of 2,3-Dimethylquinoxaline: an Antimicrobial Found in a Traditional Herbal Medicine” is worth publishing after applying some minor corrections. My comments are as following:

1. In the names and affiliations of the authors there are some mistakes that need to be corrected:

• The number of the author “Abdelbagi Alfadil13” as superscript is 13 please correct,

• There should be a comma between the second and the first author,

• The comma after the number of the 4th author “Huda Alkreathy2,” should be removed,

• There are dots after the numbers of the 5th and 6th authors which should be removed, also there should be a comma between their names.

• The font size and the and the font type should be uniform in the 1st page

2. Authors should start with general statements about the relevant subject in the introduction part,

3. Authors should extend the introduction part with explanatory and literary comments.

4. The title “2.9 Oral toxicity study with repeated doses in mice (28 days )” should be written with bold

5. Authors should write a more detailed conclusion

6. Please add the following related references in the introduction part

• Yener İsmail, Yilmaz Mustafa Abdullah, Tokul Ölmez Özge, Akdeniz Mehmet, Tekin Fethullah, Haşimi Nesrin, Alkan Mehmet Hüseyin, Öztürk Mehmet, Ertaş Abdulselam (2020). A Detailed Biological and Chemical Investigation of Sixteen Achillea Species’ Essential Oils via Chemometric Approach. CHEMISTRY & BIODIVERSITY, 17(3) e1900484, https://doi.org/10.1002/cbdv.201900484.

• Selek Şehabettin, Koyuncu İsmail, Çağlar Hifa Gülru, Bektaş İbrahim, Yilmaz Mustafa Abdullah, Gönel Ataman, Akyüz Enes (2018). The evaluation of antioxidant and anticancer effects of Lepidium Sativum Subsp Spinescens L. methanol extract on cancer cells. Cellular and Molecular Biology, 64(3), 72-80., https://doi.org/10.14715/cmb/2018.64.3.12.

• Mehmet Akdeniz, Ismail Yener, Mustafa Abdullah Yilmaz, Sevgi Irtegun Kandemir, Fethullah Tekin, Abdulselam Ertas (2021). A potential species for cosmetic and pharmaceutical industries: Insight to chemical and biological investigation of naturally grown and cultivated Salvia multicaulis Vahl. Industrial Crops & Products, 168, 113566, https://doi.org/10.1016/j.indcrop.2021.113566.

• Mehmet Akdeniz, Ismail Yener, Abdulselam Ertas, Mehmet Firat, Baris Resitoglu, Nesrin Hasimi, Sevgi Irtegun Kandemir, Mustafa Abdullah Yilmaz, Asli Barla Demirkoz, Ufuk Kolak and Sevil Oksuz (2021). Biological and Chemical Comparison of Natural and Cultivated Samples of Satureja macrantha C.A.Mey. Records of Natural Products, 15 (6), 568-584, http://doi.org/10.25135/rnp.237.21.02.1957.

34. Ebubekir Izol, Hamdi Temel, Mustafa Abdullah Yilmaz, Ismail Yener, Ozge Tokul Olmez, Erhan Kaplaner, Mehmet Fırat, Nesrin Hasimi, Mehmet Ozturk, Abdulselam Ertas (2021). A Detailed Chemical and Biological Investigation of Twelve Allium Species from Eastern Anatolia with Chemometric Studies. Chemistry and Biodiversity, 18 (1), e2000560, https://doi.org/10.1002/cbdv.202000560.

Reviewer #2: The document submitted for our consideration is a scientific work of significance, providing information on the toxicological profile of 2,3-dimethylquinoxaline (DMQ), a broad-spectrum antimicrobial derived from plants.

Abstract

In vitro and in vivo remain a group of words which are necessarily italicised.

Introduction

Chromolaenaodorata is in two words Chromolaena odorata

Please update the references in the second paragraph of the introduction. The references used are too old.

Material and methods

You were right to specify in this study that the mice used were non-pregnant females. Are they nulliparous or multiparous? These are parameters to be taken into account for the tests.

This choice must be justified

Laboratory animals, particularly females, are more sensitive than males and when females are pregnant or nulliparous, the tests may be influenced by these physiological characteristics. Here are some of the reasons why it is important to specify the physiological characteristics of the animals tested

6. PLOS authors have the option to publish the peer review history of their article (what does this mean?). If published, this will include your full peer review and any attached files.

Reviewer #1: **Yes: **Mustafa Abdullah YILMAZ

Reviewer #2: **Yes: **Lamine Baba-Moussa

---

## [Decision Letter · Decision Letter 1]

10 Jul 2024

In-Vivo and In-Vitro Toxicity Evaluation of 2,3-Dimethylquinoxaline: an Antimicrobial Found in a Traditional Herbal Medicine

PONE-D-24-06527R1

Dear Dr. Alfadil,

We’re pleased to inform you that your manuscript has been judged scientifically suitable for publication and will be formally accepted for publication once it meets all outstanding technical requirements.

Kind regards,

Ahmed E. Abdel Moneim

Academic Editor

PLOS ONE

Additional Editor Comments (optional):

Reviewers' comments:

Reviewer's Responses to Questions

**Comments to the Author**

1. If the authors have adequately addressed your comments raised in a previous round of review and you feel that this manuscript is now acceptable for publication, you may indicate that here to bypass the “Comments to the Author” section, enter your conflict of interest statement in the “Confidential to Editor” section, and submit your "Accept" recommendation.

Reviewer #2: All comments have been addressed

2. Is the manuscript technically sound, and do the data support the conclusions?

Reviewer #2: Yes

3. Has the statistical analysis been performed appropriately and rigorously? 

Reviewer #2: Yes

4. Have the authors made all data underlying the findings in their manuscript fully available?

Reviewer #2: Yes

5. Is the manuscript presented in an intelligible fashion and written in standard English?

Reviewer #2: Yes

6. Review Comments to the Author

Reviewer #2: All my concerns have been well adressed. The Paper could be published in Current form. Congratulations.

7. PLOS authors have the option to publish the peer review history of their article (what does this mean?). If published, this will include your full peer review and any attached files.

Reviewer #2: **Yes: **

---

## [Editor Report · Acceptance letter]

9 Aug 2024

PONE-D-24-06527R1 

PLOS ONE

Dear Dr. Alfadil, 

I'm pleased to inform you that your manuscript has been deemed suitable for publication in PLOS ONE. Congratulations! Your manuscript is now being handed over to our production team.

Kind regards, 

on behalf of

Dr. Ahmed E. Abdel Moneim 

Academic Editor

PLOS ONE